# Differentially Expressed Conserved Plant Vegetative Phase-Change-Related microRNAs in Mature and Rejuvenated Silver Birch In Vitro Propagated Tissues

**DOI:** 10.3390/plants12101993

**Published:** 2023-05-16

**Authors:** Baiba Krivmane, Kaiva Solvita Ruņģe, Ineta Samsone, Dainis Edgars Ruņģis

**Affiliations:** Latvian State Forest Research Institute “Silava”, 111 Rīgas st, LV-2169 Salaspils, Latviakaivarunge@gmail.com (K.S.R.);

**Keywords:** micropropagation, recalcitrant, woody plants, perennial plants, *Betula pendula*

## Abstract

In plants, phase change from the juvenile stage to maturity involves physiological and anatomical changes, which are initiated and controlled by evolutionary highly conserved microRNAs. This process is of particular significance for the in vitro propagation of woody plant species, as individuals or tissues that have undergone the transition to vegetative maturity are recalcitrant to propagation. Conserved miRNAs differentially expressed between juvenile (including rejuvenated) and mature silver birch tissues were identified using high-throughput sequencing of small RNA libraries. Expression of some miR156 isoforms was high in juvenile tissues and has been previously reported to regulate phase transitions in a range of species. Additional miRNAs, such as miR394 and miR396, that were previously reported to be highly expressed in juvenile woody plant tissues were also differentially expressed in this study. However, expression of miR172, previously reported to be highly expressed in mature tissues, was low in all sample types in this study. The obtained results will provide insight for further investigation of the molecular mechanisms regulating vegetative phase change in silver birch and other perennial woody plant species, by analysing a wider range of genotypes, tissue types and maturation stages. This knowledge can potentially assist in identification of rejuvenated material at an earlier stage than currently possible, increasing the efficiency of silver birch in vitro propagation.

## 1. Introduction

In plants, phase change from the juvenile stage is facilitated by a series of physiological and genetic changes [1]. This transition occurs in all plants, but while the particular phases are more pronounced in woody plant species, the majority of molecular genetic studies on the regulation of phase change have been performed in annual model or crop species [2]. In model species, such as *Arabidopsis thaliana*, transition from the juvenile to the mature phase is well characterized and understood at the molecular level. At a physiological level, this transition is controlled by the interaction between phytohormones such as cytokinins and gibberellins [3]. In woody and perennial species, such as trees, the phase transitions are less well understood but are thought to involve similar hormonal signalling pathways. Additionally, environmental factors such as temperature, light, and water availability play a role in the timing of phase transitions [4,5]. Clonal propagation of woody perennial plant species is often difficult, particularly if individuals or tissues have transitioned from the juvenile phase to vegetative maturity. The evolutionary highly conserved microRNAs (miRNAs) miR156 miR172 and their target genes have been shown to be involved in the transition from juvenility to maturity in a number of models and annual plant species [6,7]. Similar regulatory pathways and genes have also been associated with vegetative phase change in some woody plant species [8,9,10,11,12]. The molecular mechanisms regulating phase change involve *SQUAMOSA* promoter binding protein-like (SBP/SPL) transcription factor genes, which are targeted by miR156, promoting the transition to maturity by repressing the expression of *SPL* genes, and consequently upregulating key MADS-box genes that induce flowering, such as *APETALA (AP1), LEAFY (LFY) and FRUITFULL (FUL)* [13]. MiR156 suppresses the expression of miR172 through interactions with the *SPL9* and *SPL10* genes. This results in increased expression of miR172-targeted *AP2*-like transcription factor family genes such as *APETALA2* (*AP2*), *TARGET OF EAT1* (*TOE1*), *TOE2* and *TOE3*, *SCHLAFMUTZE* (*SMZ*), and *SCHNARCHZAPFEN* (*SNZ*) [14]. The two most studied miRNAs involved in phase transition are miR156 and miR172. Expression of miR156 is high in juvenile tissues, with expression decreasing during the transition to maturity. Conversely, expression of miR172 is low in juvenile stages, increasing with maturity. However, this increase in miRNA172 expression in mature tissues is more universally observed in annual species and is not always observed in some perennial, woody species [9].

MiRNAs are short (usually between 20 and 24 nucleotides in length), nonprotein-coding RNAs, which post-transcriptionally regulate gene expression, either by cleavage or by translational repression via complementary binding to messenger RNA (mRNA) [15]. Primary microRNA transcripts (pri-RNAs) are one to several hundred nucleotides in length and contain at least one hairpin-stem loop. Enzymes such as Dicer-like enzyme 1, Hasty and others cleave these primary transcripts into approximately 70 nt long precursor miRNAs (pre-miRNAs), which are subsequently cleaved into ~20–24 nt long mature miRNA molecules [16]. Based on the sequence homology of mature miRNA molecules, miRNAs are categorized into families, consisting of miRNA isoforms or isomiRs, which can have minor sequence or length differences, but which have the same or similar target genes. However, the primary and precursor sequences that encode identical or highly similar mature miRNAs can have differing sequences surrounding the mature miRNA site, and the precursors may be differentially expressed in response to different abiotic or biotic conditions, or at various developmental stages or tissues [17]. Due to their short length, one miRNA could interact with a number of different mRNAs [18].

Silver birch (*Betula pendula* Roth.) is widely distributed in northern European boreal forests and is a significant tree species from both ecological and economic aspects. Forest tree breeding programs, producing productive and high-quality improved silver birch reproductive material have been developed in northern European countries [19,20]. Breeding programs can be made more efficient, and the time required for controlled crossings, evaluation and deployment of improved material can be reduced by up to 10–15 years by the clonal reproduction of high-quality genotypes [21]. In specific conditions, plantations established using clonally propagated silver birch genotypes that have been selected for particular traits can have higher productivity than stands established from seed orchard material [22]. To obtain the most accurate assessment of tree productivity and quality, phenotypic evaluation of silver birch is often undertaken when trees have reached their mature phase. Therefore, in order to successfully propagate this material in vitro, the explants must undergo a rejuvenation process. Mature plant tissues can be rejuvenated by a number of methods, including in vitro culturing [14]. Variations in growth media composition or other in vitro conditions can be used to successfully rejuvenate mature tissues. Some birch genotypes can be rejuvenated and regenerated in vitro, while others persist in a mature state and progressively senesce and cannot be successfully propagated [21,23]. The rejuvenation process during the in vitro propagation of woody plant species has not been widely studied, and no information is available about the functions and the expression patterns of miRNAs and target genes involved in phase change in silver birch. Knowledge about the molecular basis of phase change in silver birch could allow for the more efficient identification of rejuvenated explants that are able to be propagated, as well as in vitro cultivation conditions that facilitate the rejuvenation of mature explants. A silver birch (*Betula pendula* Roth.) reference genome is available [24], and there is one published report on the identification of miRNAs from silver birch pollen, but this was in the context of immune and autophagic responses in human lung cells [25].

Information about vegetative phase status can assist in the development of in vitro protocols and in selecting the optimum explants and growth conditions. Currently, rejuvenated explants are identified by successful in vitro propagation, which can take many months and requires the maintenance of a large number of cultures, many of which will not rejuvenate and will not be able to be propagated in vitro. Attempts have been made to use phenotypic or anatomic traits (e.g., leaf and stem morphology, cellular and subcellular anatomy) to differentiate juvenile and mature tissues in in vitro culture; in a number of species, however, these differences are often subtle and difficult to unambiguously assess and can also take a long time to develop [8,26,27]. In addition, there are often variations in morphological and propagation efficiency between and within genotypes, even if cultured in identical conditions. As vegetative phase change is coordinated at a molecular level by the differential expression of genes and by the regulation and attenuation of gene expression (e.g., via microRNAs), investigation of the regulatory elements controlling this process can provide a more accurate assessment of the juvenility of explant material and the effect of in vitro culture conditions on the rejuvenation of tissues. Identification of miRNAs involved in the rejuvenation process could provide the opportunity to identify rejuvenating explants at an earlier stage, increasing in vitro propagation efficiency.

In this study, conserved miRNAs that were differentially expressed between rejuvenated and mature silver birch in vitro explants were identified using high-throughput sequencing of small RNA libraries. Seedlings and mature leaves were used as juvenile and mature controls, respectively. The differentially expressed miRNAs identified in this study can be utilized to assess the rejuvenation of silver birch explants in in vitro cultures. Early identification of rejuvenated in vitro cultures can increase the efficiency of vegetative propagation by the removal of unrejuvenated material, and it can assist in the optimization of in vitro culture conditions. In addition, the obtained results will provide a basis for further investigation of the molecular mechanisms regulating vegetative phase change in silver birch and other perennial woody plant species, by utilizing the obtained results to analyse a wider range of genotypes, tissue types and maturation stages.

## 2. Results

### 2.1. Identification of Conserved miRNAs in Silver Birch

A total of 15 small RNA libraries were sequenced from four sample types—mature in vitro shoots (IVM) (four libraries), rejuvenated in vitro shoots (REV) (four libraries), leaves from a mature (approximately 20 years old) silver birch (MAT) (three libraries) and 3-week-old seedlings (JUV) (four libraries). A total of 37.9 million reads were obtained, of which 8.2 million reads were without barcode information (28%). A total of approximately 29.7 million barcoded reads were obtained from the 15 libraries (an average of 2.0 million reads per library). Of the 2.2 billion bases, 1.9 billion had a quality score of more than Q20. After trimming of sequences by length (minimum length 18 nt and maximum length 25 nt), 14.87 million reads remained, which were clustered into 2,776,807 unique sequences with a minimum read count of five sequences over all libraries. Comparison of these unique sequences with the miRBase v22 database identified 2600 conserved miRNA isomiR sequences, which were assigned to 291 miRNA groups. These 291 miRNA groups belonged to 50 different miRNA families. Of the 2600 conserved miRNA sequences, 1932 were ambiguously annotated, i.e., a small RNA sequence was similar to the mature regions of two different miRBase sequences (from the same miRNA family). Of the conserved miRNA sequences identified in our data set, 43.27% were identified by homology with Glycine max sequences, 16.55% with Populus euphratica sequences and 7.92% with Oryza sativa sequences. Only 0.01% had homology with Acacia magnum sequences and 0.23% with Pinus densata sequences (Appendix A).

The largest number of different miRNA families was found in mature in vitro (IVM) samples (46 families), and in rejuvenated (REV) samples (45 families), and the smallest number were found in mature control (MAT) samples (33 families). The majority of isomiRs were found for the miR166 family with 470 isomiRs, as well as the miR159 family with 376 isomiRS, and the miR156 family with 314 isomiRs. The highest number of isomiRs was found in IVM samples (1291) followed by JUV samples (1183), and the smallest number was found in MAT samples (645). Of the 50 conserved miRNA families identified, only one family (miR395) was not found in IVM samples, and three families (miR2950, miR9726 and miR1507) were not found in REV samples (Appendix A). The most highly expressed isomiRs in all sample types were from the miR166 (TCGGACCAGGCTTCATTCCCC and TCTCGGACCAGGCTTCATTCC) and miR156 (TTGACAGAAGATAGAGAGCAC) families.

The miR156 family was found in all sample types, with the highest read count (696) in JUV samples and lowest (28) in MAT samples. The miR172 family was found in all sample types, but with very low expression levels—in juvenile control (JUV) samples the expression value was 1, 7 in MAT samples, 1 in REV samples and 2 in IVM samples. Only nine isomiRs were found for the miR172 family.

Four miRNA families (miR947, miR950, miR951 and miR3711), which were previously reported as conifer-specific miRNAs [28], were also found in this study, but with low expression values (read counts)—9, 15, 31 and 45. Of these, miR947 was not found in JUV samples. miR950 was found in all sample types, but miR951 was found only in REV and IVM samples. miR3711 was not found in MAT samples.

### 2.2. Differentially Expressed miRNAs

The identified conserved miRNAs were analysed to identify significantly differentially expressed miRNAs between the four sample types (REV, IVM, JUV, MAT). miRNAs showing a fold change in expression of ≥1.5 or ≤ −1.5 and an FDR-corrected *p* value < 0.05 were deemed as differentially expressed.

Comparing MAT and JUV samples, 19 isomiR groups were differentially expressed (8 downregulated and 11 upregulated in JUV samples): three isomiR groups from each of the families miR156 and miR408, two isomiR groups from each of the miR165, miR166, miR169 families, and one isomiR group from each of the miR171, miR394, miR396, miR398, miR472, miR482, miR858 families (Figure 1). IsomiR groups from the miR166 and miR165 families were highly expressed in MAT samples, intermediately expressed in IVM and REV, (with lower expression in REV samples) and had low expression in JUV samples. Opposite expression patterns were observed for miR169 isomiR groups, which were highly expressed in JUV samples, intermediately expressed in REV and IVM samples (with lower expression in IVM samples) and had low expression in MAT samples (Figure 1; Appendix A). Two miR156 isomiR groups had higher expression in JUV samples and lower expression in MAT samples, with similar expression patterns between the REV and IVM samples, i.e., higher expression in REV samples than in IVM samples. The third differentially expressed miR156 isomiR group was also highly expressed in JUV samples and lowly expressed in MAT samples, but higher expression was found in IVM samples compared to REV samples.

Five isomiR groups were differentially expressed between IVM and REV samples, all of which were more highly expressed in IVM samples (Figure 2). Of these, four were more highly expressed in MAT and IVM samples but had lower expression in JUV and REV samples (two miR858 isomiR groups, one miR159 isomiR group and one miR8175 isomiR group). One of the miR858 isomiR groups was also statistically significantly differentially expressed between MAT and JUV samples, as well as IVM and JUV samples, and the miR8175 isomiR group was differentially expressed between REV and MAT samples.

The comparisons that could most likely identify miRNAs associated with vegetative phase change in silver birch were the MAT-JUV and IVM-REV comparisons described above. However, differentially expressed miRNAs were identified in all pairwise comparisons between sample types. A total of 41 differentially expressed miRNA isomiR groups were differentially expressed between the sample types (Appendix A).

Fourteen isomiR groups were differentially expressed between REV and JUV samples. Two isomiR groups each from the miR169 and miR403 families, and one isomiR group from each of the miR396, miR5368, miR5077 families had higher expression in the JUV samples and seven, miR157, miR166, two of miR171 isomiR group, miR472, miR482, and miR2118, were more highly expressed in REV samples.

Seventeen miRNA isomiR groups were differentially expressed between IVM and JUV samples, twelve more highly expressed in IVM samples (two miR171 groups, two miR858 isomiR groups, and one isomiR group from each of the miR157, miR166, miR167, miR319, miR472, miR482, miR2118, miR3627 families), and five (two miR169 isomiR groups, one isomiR group from each of the miR394, miR395, miR396 families) were more highly expressed in JUV samples. Of these, two miRNA isomiR groups, miR169 and miR394, were also highly expressed in IVM samples, but the miR482 isomiR group was highly expressed in JUV samples. Only three isomiR groups (miR169, miR394 and miR396) had the expected expression pattern—higher expression in JUV, lower in MAT and IVM samples.

Eleven miRNA isomiR groups were differentially expressed between REV and MAT samples, ten more highly expressed in REV samples, and one more highly expressed in MAT samples. From these families, four miR156 isomiR groups, two miR408 isomiR groups, one miR157 isomiR group, one miR403 isomiR group, one miR8175 isomiR group, and one miR2118 isomiR group were downregulated, and one miR169 isomiR group was upregulated in MAT samples.

Ten miRNA isomiR groups were differentially expressed between the IVM and MAT samples, nine more highly expressed in IVM samples (three miR156 and miR408 isomiR groups, one isomiR group from each of miR157, miR171, miR3711), and one more highly expressed in MAT samples (one miR169 isomiR group).

### 2.3. Identification of Potential miRNA Precursors for Differentially Expressed miRNAs

Mapping of the differentially expressed mature miRNA sequences to the *Betula pendula* scaffold assembly, vv1.2 scaffolds (University of Helsinki id 35079), identified 55 potential precursor sequences for 35 isomiR groups (20 miRNA families, 623 isomiRs) from a total number of 41 differentially expressed isomiR groups (708 isomiRs) (Appendix A). The most precursors (20) were found for the miR156 family, and only one potential precursor sequence was found for the miR171, miR319, miR408, miR2118, miR3627, miR5368 families. No potential precursors were found for the differentially expressed miR159, miR858, miR3711, miR5077, miR8175 family isomiR groups. Homologous sequences in the *Betula pendula* scaffold assembly were found for some (miR156, miR166, miR167, miR169, miR171, miR319, miR403, miR5368) isomiR groups, but these had a minimum free-folding energy index (MFEI) of less than 0.85 and therefore were not considered as miRNA precursors. A number of sequences were identified as potential precursors for multiple miRNA isoform groups from the same family. This was due to the parameters utilized, which allowed for a maximum of two mismatches between a miRNA and a potential precursor miRNA sequence.

Different precursors on the same silver birch contig were found for miR403, miR408 (Contig129) and for miR156, miR165 (Contig2096). For both miR3627 and miR395, three and four repeated precursor sequences were found on one contig, respectively. This suggests the clustering of the miRNA precursor sequences in the silver birch genome.

Potential precursor sequences for conserved miRNAs were analysed for the ability to form the required stem–loop structure, and the minimal folding free energy indexes were calculated. Predicted pre-miRNA sequences were trimmed in the primary miRNA sequence region until the next bulge or loop after the miRNA* region. Minimum folding free energy indexes ranged from 0.48 to 1.32, with most being >0.85 (45 precursor sequences (82%)) (Appendix A).

### 2.4. Expression Analysis of Precursor miRNAs Using qPCR

Primers for qPCR were designed for 19 of the identified precursor sequences (Appendix A). The precursors were selected based on their differential expression between MAT-JUV and/or IVM-REV groups. As described above, precursor sequences were not identified for all differentially expressed miRNAs. Expression of these selected precursors, as well as precursors for the miR156 and miR172 families, was determined as described previously [29]. Relative expression between the sample group MAT, IVM, REV and JUV was determined for 11 precursors, each from a different miRNA family (Appendix A). The remaining primer pairs either did not amplify a PCR product, or amplified non-specific fragments, and therefore, further optimisation or redesign of the PCR primers is required.

The relative expression patterns of the precursors miR156_789, miR394a_134, miR396g_126, miR408_129 were the same as those of their corresponding mature miRNAs obtained from the sequencing data. For all of these precursor and mature miRNAs, expression levels were lowest in the MAT samples. For the precursors miR171h_386 and miR482_376, the relative expression patterns were similar to the sequencing data, with the exception of the JUV samples, which had high expression in the qPCR analysis, but expression of the mature miRNA from sequencing data was low. Similarly, relative expression of the precursor miR398_485 was similar to the sequencing data for the mature miRNA, except for the JUV samples, which had low expression in the qPCR analysis, but expression of the mature miRNA from sequencing data was high. Relative expression of the precursors miR166h_50, miR169a_800 and miR472_1354 was not consistent with the expression of the mature miRNAs from sequencing data. Relative expression of the miR166h_50 precursor was low in the MAT and IVM samples and similar in the REV and JUV samples, while from the sequencing data expression of the miR166 family, it was high in the MAT samples and lower in the other samples. Relative expression of the precursor miR472_1354 was similar in the IVM, REV and JUV samples, while expression in the MAT samples was increased only 2.5-fold, while from the sequencing data, expression of the mature miRNA was lowest in the JUV samples and highest in the IVM samples. For the precursor miR169a_800, the errors associated with the RQ values were large for all analysed samples, and the MFEI for the precursor sequence was low (0.74); therefore, this may not be the correct precursor sequence for the miR169 family. Relative expression patterns for the precursor miR172_1931 were as expected from other studies, with high expression in the mature samples (MAT and IVM) and low expression in the juvenile samples (JUV and REV). However, from the sequencing data, expression of mature miRNAs from the miR172 family was extremely low in all samples, including the mature samples (MAT and IVM). In general, the most consistent comparisons of expression of the precursor miRNAs by qPCR and mature miRNAs by sequencing were for miRNAs that were more highly expressed in juvenile tissues.

### 2.5. miRNA Target Gene Identification

Analysis of putative target genes identified 757 genes targeted by 3740 isomiRs from the 19 miRNA groups that were differentially expressed between the MAT and JUV samples (Appendix A). The predicted target genes included stress-related gene families such as *Leucine Rich Repeat (LRR)* protein genes (miR156, miR398), protein kinase domain (miR408), *TIR-NBS* or *TIR-NBS-LRR* type disease resistance protein (miR472 and miR482), heat shock protein (miR156 and miR482), and blue copper-like protein (miR408) genes. Target transcription factors included genes such as *SBP* (miR156), *MYB* (miR858, miR166), *HD ZIP III* (miR165 and miR166), zinc finger protein genes, superoxide dismutase (Cu-Zn) (miR156 and miR398) and others. Target genes also included transferases such as glutathione S-transferase (miR169), acetyltransferase, GNAT family protein (miR408) and DNA methyltransferase (miR166).

Twenty-two of the identified putative target genes (targeted by one miRNA group—bpe-miR166-3p) were associated with malate dehydrogenase or NAD-dependent malate dehydrogenase. Twelve of the identified putative target genes (targeted by three miRNA groups—bpe-miR482-3p, bpe-miR472-3p, bpe-miR166-3p) were associated with NBS-LRR, TIR-NBS-LRR and TIR-NBS type disease resistance proteins. Plant NBS-LRR proteins are involved in the detection of pathogen-associated proteins, most often the effector molecules of pathogens, which are responsible for virulence. Eight targets for three miR156 groups with 29 isomiRs were associated with squamosa promoter binding-like proteins. One gene targeted by bpe-miR166a-3p was maturase, which are prokaryotic enzymes that aid self-excision of introns in precursor RNAs and have evolutionary ties to the nuclear spliceosome. Five miR169 isomiRs targeted extracellular ligand-binding receptor precursors. The miRNA group bpe-miR858b targeted four different MYBPA1 protein genes.

Analysis of conserved miRNA putative target genes of the five isomiR groups differentially expressed between IVM and REV samples identified 195 genes (Appendix A). The predicted target genes included MYB and MYBPA1 transcription factors, which were targeted by bpe-miR858b and bpe-miR858-5p isomiR groups. Bpe-miR159a targeted a number of genes, including AUX/IAA protein gene, DNA-directed RNA polymerase genes (two target genes), NBS-LRR type or TIR-NBS disease-resistance protein genes (two target genes).

## 3. Discussion

In vitro vegetative propagation of silver birch is complicated by the fact that explants are often obtained from mature tissues and therefore must be rejuvenated by prolonged maintenance under in vitro culture conditions. Identification of rejuvenated samples by morphological and anatomic parameters can be variable and unreliable, and they can take a long time to develop and evaluate. This study identified miRNAs that were differentially expressed between juvenile (rejuvenated) silver birch tissues and mature tissues. Higher expression of members of the miR156 family was observed in juvenile samples, confirming the role of this miRNA family in juvenility [7]. From the sequencing results, the expression of the miR172 family was extremely low in all sample types, and none of the identified miR172 transcripts had significantly higher expression in mature samples. However, the relative expression of miR172 precursor sequences determined by qPCR was higher in mature samples. Previous studies of the expression of the miR172 family in other woody tree species and correlation with plant maturity have reported differing results. Expression of miR172 was not significantly different between juvenile and mature tissues in macadamia and mango, but it was higher in mature avocado tissues [9]. However, higher expression of miR172 in mature tissues has also been reported in other woody plant species [8,30]. Most studies of miR172 expression in phase transition have been conducted in annual species, where vegetative phase change is often difficult to separate from the transition to flowering [2], and many studies in woody tree species investigated transition to flowering. In studies of perennial species investigating traits other than flowering, the expression of miR172 was not as well-correlated with other developmental traits, e.g., rooting in *Eucalyptus* [31] and seed germination in *Nelumbo nucifera* Gaertn. [32]. In this study, the explants taken for establishment of in vitro cultures were closed buds and were not obviously transitioning to flowering. This could explain why the expression of miR172 was low and not significantly different between juvenile and mature sample types. However, further studies on the expression of both mature and precursor miR172 sequences is required to determine the role of this miRNA family in phase transition in silver birch.

Other significantly differentially expressed conserved miRNAs identified in this study included miR394 and miR396, both of which were more highly expressed in juvenile tissues and which were also upregulated in juvenile avocado, mango and macadamia leaves [9]. MiR394 regulates processes in the shoot apical meristem (SAM) [33] and regulates flowering time in *Arabidopsis thaliana* [34]. The miR396 family repress expression of the transcription factor class GROWTH REGULATING FACTOR (GRF) and GRF-INTERACTING FACTOR (GIF) transcriptional co-regulators, and this complex is crucial in many aspects of plant growth and development in angiosperms [35]. A number of the differentially expressed miRNAs are ancient and highly conserved, suggesting their central role in plant development. MiR398 is extremely ancient and is found in both gymnosperms and angiosperms. This miRNA has been shown to be involved in growth and development processes in a number of plant species, as well as regulating abiotic and biotic stress responses [36]. Other ancient and highly conserved microRNAs include miR165/miR166, which regulate HD-ZIP III (homeodomain-leucine zipper) transcriptional factors and are involved in a range of plant development processes [37,38]. MiR171 is highly conserved in land plants, is found in many lineages, from bryophytes to angiosperms, and regulates shoot meristem activity and phase transition [39]. MiR408 plays a significant role in regulating plant growth, development and stress response in a range of plant species [40]. Many of these highly conserved miRNA families have been shown to play a role in very diverse processes, as they interact with transcription factor gene families. The miR858 family targets the large *MYB* families that regulate numerous biological processes from plant development to stress responses and secondary metabolism pathways [41].

Other differentially expressed miRNAs are less well studied and thus currently have been reported to be involved in specific processes, according to experimental design; however, further studies may show that they are also involved in other plant developmental aspects. MiR482 targets *NBS-LRR* genes and is involved in plant resistance mechanisms [42], and miR8175 is upregulated in aluminium tolerance responses in plants [43]. MiR5077 is reported to be involved in sugar metabolism in pear fruit development [44], which could play a role in vegetative phase transition, as sugars have been shown to be a cue for this transition [5].

This study identified several ancient and conserved miRNA families that were differentially expressed between the juvenile and mature samples analysed, and which have been previously described to be involved in phase transition in a wide range of species (e.g., miR156, miR394, miR408). These were miRNAs that are more highly expressed in juvenile samples. Other miRNAs that have been previously reported to be more highly expressed in mature tissues were not upregulated in mature tissues in this study. Expression of miR396 was increased in mature *Sequoia sempervirens* tissues [45], while in this study, expression of a member of the miR396 family was highest in the juvenile control samples and lowest in the mature control samples (Appendix A). Expression of miR172 is widely reported to be increased in mature tissues in a wide range of annual and perennial species; in this study, expression of mature miR172 sequences was very low in all samples. However, analysis of relative expression of precursor miRNAs, using real-time PCR in this study and in a previous study [29], indicated that expression of two miR172 precursors was increased in mature silver birch leaves. As mentioned previously, as the majority of previous studies of miRNA expression and phase change have been conducted in annual species, increased expression of miR172 may be more related to the transition to flowering, which is not as differentiated from vegetative phase change in annual species compared to perennial species. In addition, the timing of vegetative phase change in trees is extremely variable [8], and the differing results from different studies in tree species may be a reflection of this heterogeneity. Due to the more discrete separation of vegetative phase change and transition to flowering in perennial woody species, further investigations are needed to clarify the role of these conserved miRNAs in tree species. Some of the differentially expressed miRNAs identified in this study have been reported to be involved in the regulation of flowering time in *Arabidopsis thaliana* (e.g., miR394) [34], but additional studies in tree species are required to determine if these miRNAs are also involved in the transition to flowering or if different isoforms or miRNA families are involved.

Comparison of the expression patterns of selected precursor miRNAs using qPCR with the sequencing data generally revealed that the most consistent patterns of expression were for miRNAs that were more highly expressed in juvenile tissues. This may be due to the fact that in this study, rejuvenated in vitro samples were defined by their ability to be propagated and thus had reverted to a juvenile state, while the mature in vitro samples may be at different stages along the spectrum from juvenility to maturity. More generally, plant juvenile stages are more strictly defined than maturity, as plant maturity can encompass a range of life stages, e.g., vegetative maturity and reproductive maturity, and these stages can be overlapping in time and within individuals. In addition, expression analysis of mature miRNAs and precursor miRNAs provide complementary information, as different precursor sequences may encode the same or highly similar mature miRNAs, but the precursors could have differing expression patterns at different developmental stages or tissues, or in response to different abiotic or biotic conditions [17].

Further analysis of the sequencing data will potentially enable identification of novel miRNAs that are differentially expressed between juvenile and mature silver birch tissues. Identification of precursor sequences and in silico analysis of the folding structures can indicate if the expected stem–loop hairpin structures can be formed. Initial results about the expression of the precursor miRNAs using real-time PCR indicate that the miRNAs with increased expression in juvenile tissues are more consistent with the sequencing results of mature miRNAs. Further research on the differential expression of precursor miRNAs at different time points or tissues will provide information on the specific expression of precursor sequences that encode the same or highly similar mature miRNAs. In addition, transcriptome analysis will provide an opportunity to compare expression of miRNAs with their potential target gene expression levels. The expressions of the miRNAs identified in this study need to be examined in a wider range of germplasm to determine the effect of genetic background on miRNA expression and to examine variation of miRNA expression of different clones with similar in vitro propagation properties.

## 4. Materials and Methods

### 4.1. Plant Material and Extraction of RNA

In vitro samples were divided into two groups: rejuvenated shoots which were able to be propagated, and mature shoots which did not proliferate and which exhibited typical signs of maturity. The in vitro samples were derived from two different clones (VKA and 54–257), but both rejuvenated and mature shoots were obtained from both clones, and therefore, samples were analysed according to juvenility/maturity status rather than by clonal identity. These clones are derived from the silver birch breeding program and were not selected by any particular criteria other than the morphogenic properties of the in vitro shoots derived from them. Plant material for miRNA expression analysis was collected and stored at −80 °C until RNA extraction. Total RNA was extracted from leaves of four rejuvenated in vitro shoots (REV samples) and four mature in vitro shoots (IVM samples). IVM cultures from the clone VKA exhibited typical signs of maturity (thick stems, large and thick leaves, inability to proliferate). IVM cultures from clone 54–257 exhibited the previously mentioned typical signs of maturity, as well as having partially yellow leaves. All in vitro samples had been maintained in in vitro culture for approximately 10 months when collected for analysis. Photographs representative of the in vitro samples analysed are included in Appendix A. Leaves from a mature (approximately 20 years old) silver birch (3 samples, clone VKA) were used as a mature control (MAT), and 3-week-old seedlings (4 samples, seeds collected from clone VKA) were used as a juvenile control (JUV). Samples for the MAT control were collected in three different months—on 21 May, 10 July and 3 September 2018, but seeds for JUV control seedlings were collected in November 2019 and grown in turf substrate for 3 weeks. RNA was extracted using a standard phenol/chloroform/isoamyl alcohol protocol [46]. Total RNA, extracted from all samples, was treated with the Turbo DNA-free kit (Ambion by Life Techologies, Carlsbad, USA, Cat. No. AM1907) following the manufacturer’s instructions.

### 4.2. RNA Quality Control

RNA concentration was measured with a Qubit and QuantiT™ RNA BR Assay Kit (Thermo Fisher, Eugene, USA, Cat. No. Q10210). RNA purity (DNA contamination) was tested by polymerase chain reaction (PCR) with the extracted RNA solution as template and three birch genomic microsatellite locus primer pairs L7.8, L7.4 and L1.10 [47]. Each forward primer was labelled with a different fluorophore (6-FAM, HEX, or TMR) to facilitate visualization using capillary electrophoresis. The PCR reactions for the microsatellite markers were carried out in a 10 μL solution containing a final concentration of 1 × HOT FIREPol^®^ Blend Master Mix with 10 mM MgCl_2_ (Solis Biodyne, Tartu, Estonia, Cat. No. 04-27-00120), 0.3 mM of each primer, 1 μL RNA solution. PCR cycling conditions consisted of an initial denaturation step of 95 °C for 15 min, 35 cycles of 95 °C for 20 s, 55 °C for 30 s, and 72 °C for 45 s, followed by a final extension step of 72 °C for 10 min. PCR reactions were carried out in an Eppendorf Mastercycler gradient thermal cycler. Amplification fragments were separated on an ABI Prism 3130xl Genetic Analyzer (Applied Biosystems, Warrington, UK) and genotyped with GeneMapper 3.5 (Applied Biosystems, Warrington, UK). If no PCR amplification fragments were detected, RNA samples were considered free of DNA contamination. If PCR amplification fragments were detected, RNA samples were repeatedly treated with the Turbo DNA-free kit, and RNA concentrations and purity were reanalysed prior to reverse transcription and real-time PCR analysis. RNA purity was measured with a Qubit and QuantiT™ dsDNA HS Assay Kit (Thermo Fisher, Eugene, USA, Cat. No. Q33120). Total RNA and small RNA quality, quantity and integrity number (for total RNA) were also verified using the Agilent Technologies 2100 Bioanalyzer with the RNA Agilent RNA 6000 Nano Kit and Agilent Small RNA kit (Agilent Technologies 2100 Bioanalyzer (Agilent Technologies, Waldbronn, Germany, Cat. No. 5067-1511 and 5067-1548). Total RNA preparations were stored at −80 °C until further analysis.

### 4.3. Small RNA Enrichment, Library Preparation and Sequencing Analysis

Total RNA samples were enriched for small RNA as outlined in the Ion Total RNA-Seq Kit v2 for Small RNA Libraries Preparation guide (ThermoFisher Scientific Manual 4475936 revision B.0), and 15 small RNA barcoded libraries were prepared using the CleanTag™ Small RNA Library Preparation Kit (TriLink Biotechnologies, San Diego, CA, USA, Cat. No. L-3206) according to the manufacturer’s protocol. Each amplified sRNA library was quantified and the quality analysed using the Agilent Technologies 2100 Bioanalyzer with a High Sensitivity DNA Kit (Agilent Technologies, Waldbronn, Germany, Cat. No.: 5067-4626). Template-positive Ion Sphere™ Particles (ISPs) were prepared and enriched using the ExT Kit (Thermo Fisher, Eugene, OR, USA, Cat. No. A30670 revision E.0) on the IonChef Instrument (Ion Torrent by Thermo Fisher Scientific, Eugene, OR, USA, Cat. No. 4484177) following the manufacturer’s protocol. The enrichment was then assessed using the Qubit^®^ 2.0 Fluorometer with a dsDNA BR Assay kit. The prepared libraries were sequenced using two Ion 530 chips (Ion Torrent by Life Technologies, Cat. No. A27764) on an Ion GeneStudio™ S5 System (Ion Torrent by Thermo Fisher Scientific, Eugene, OR, USA). The sRNA sequences were analysed using the CLC Genomics Workbench software version 21.0.5 (QIAGEN, Redwood City, CA, USA). Low-quality reads and adapter sequences were removed and sequences were filtered by length for miRNA identification: minimum length 19 nt and maximum length 25 nt.

### 4.4. Conserved miRNA Identification

To identify conserved miRNAs expressed in silver birch, all unique small RNA sequences were compared to annotated plant miRNAs. Sequences from 16 species in miRBase were utilized for comparison, of which eight were tree species, including seven woody species: three conifer species, *Pinus taeda*, *Pinus densata*, *Picea abies*; four broadleaf tree species, *Populus trichocarpa*, *Populus euphratica, Acacia auriculiformis, Acacia mangium*; and one fruit tree, *Prunus persica*; as well as other plant species, *Arabidopsis thaliana*, *Arabidopsis lyrata*, *Glycine max*, *Oryza sativa*, *Nicotiana tabacum*, *Vitis vinifera*, *Zea mays, Brassica rapa*. Mature miRNA and pre-miRNA sequences of these species were obtained from miRBase (v22) and compared to unique silver birch small RNA sequences using the CLC Genomics Workbench software version 21.0.5 (QIAGEN). The silver birch unique small RNA sequences were assigned to miRNA groups, based on similarity to mature miRNA sequences from miRBase, creating consensus sequences for the identified conserved silver birch miRNAs. Two nucleotide mismatches were allowed between silver birch unique small RNA sequences and annotated plant miRNAs in miRBase, as well as two additional or missing upstream or downstream nucleotides.

Potential precursor and target gene sequences were identified for the differentially expressed conserved mature miRNA sequences. To identify potential silver birch precursor miRNA sequences, miRNA sequences were aligned to the *Betula pendula* scaffold assembly, vv1.2. (id 35079) (available from Genome evolution database at https://genomevolution.org (accessed on 1 October 2022). Small RNA sequences were mapped to silver birch sequences (contigs) using the CLC genomics workbench software Multi Blast tool (allowing two nucleotide mismatches, gaps—one nucleotide mismatch, extension—two nucleotides). Previously described criteria [28] were used with some modifications: minimum length of the double stranded segment within the folded sequence—19 nucleotides; minimum length of mature/star miRNA—19 nucleotides; maximum length of mature/star miRNA—25 nucleotides.

In addition, the minimum free-folding energy index (MFEI) was calculated to confirm that the precursor sequences conformed to the requirements for forming the miRNA precursor structures. Potential miRNA precursor sequences with a maximum of two mismatches with the mature miRNA sequences were identified and taking into account reports that plant pre-miRNAs vary from approximately 80–200 nt in length, regions flanking the mapped mature miRNAs (approximately 100 nt downstream and 100 nt upstream) were used to predict folding structures using the UnaFold program (http://www.unafold.org/mfold/applications/rna-folding-form.php (accessed on 15 November 2022) web server and the CLC genomics workbench software. If the length of a sequence was less than 200 nt, the entire available sequence was used to predict folding structures. MFE (minimal negative folding free energy, ΔG), AMFE (adjusted MFE), MFEI (minimal folding free energy index), length of sequence, nucleotide percentage (A, U, G, and C), A + U content, G + C content, and number of base pairs were calculated. The MFEI was calculated using the formula:MFEI = [(MFE/(length of the RNA sequence × 100)]/[(G + C) %]

Predicted secondary structures of precursor miRNAs have folding free energy indexes (MFEIs) ≥ 0.85, distinguishing them from other RNAs such as tRNAs and from rRNAs whose MFEI are between 0.59 and 0.66.

### 4.5. Identification of Differentially Expressed miRNAs

The identified conserved miRNAs were analysed to identify significantly differentially expressed miRNAs between the four sample types (REV, IVM, JUV, MAT). miRNAs showing a fold change in expression of ≥1.5 or ≤ −1.5 and an FDR-corrected *p* value < 0.05 were deemed as differentially expressed.

### 4.6. Target Gene Identification

Identification of potential miRNA target genes was conducted by searching for complementary regions between the mature miRNAs identified in this study and the DFCI poplar gene index (PplGI, version 5, released on 16.04.2010) using the psRNATarget-Plant Small RNA Target Analysis Server [48]. The default parameters were used for target gene identifications (seed region 2–13 nucleotides, 2 mismatches allowed in seed region, bulges (gaps) allowed, expectation = 5).

### 4.7. Analysis of Expression of Precursor miRNAs Using qPCR

Analysis of expression of selected precursor miRNAs using qPCR was conducted as described previously [29]. Primers for selected pre-miRNA amplification were designed using Primer 3 version 0.4.0. Total RNA (1 μg) was reverse transcribed using the Taqman Reverse transcription kit using an Oligo d(T)_16_ primer (Thermo Fisher Scientific, Eugene, USA, Cat. no. 4304134). The obtained cDNA was diluted to 10 ng/μL with nuclease-free water, and 2 μL of cDNA was used for qPCR analysis. Comparative Ct RT-PCR was performed with the Maxima SYBR Green/ROX qPCR Master Mix (2X) (Thermo Fisher Scientific, Eugene, USA, Cat. No. K0221) using a standard protocol on a StepOnePlus thermocycler (Thermo Fisher Scientific, PN 4376785), with three technical replications.

Two reference genes (endogenous controls) were used: actin [49] and peptidyl-prolyl isomerase (cyclophilin) [50]. Relative expression levels (relative target quantity—RQ) were determined using the 2^−ΔΔCt^ method after normalization by comparison with the endogenous control Ct values. The minimum and maximum RQ values indicate the error associated with the RQ value for the analysed precursor miRNAs and target genes. These values were computed using RQ_min_ = 2 − (RQ − SE), RQ_max_ = 2 − (RQ + SE), where SE is the standard error for the RQ. Statistical significance of differences in the expression levels of the analysed precursor miRNAs and target genes was established using one-way analysis of variance (ANOVA) and t tests, with a *p* value threshold of 0.05.

## 5. Conclusions

This initial investigation of changes in miRNA expression during vegetative phase change in silver birch confirmed numerous previous reports about the upregulation of conserved miRNAs such as miR156, miR394, miR408 in juvenile plant tissues. The results about upregulated miRNAs in mature tissues were not as equivocal; however, this has also been previously reported in tree species and may be related to the increased spatial and temporal variation of phase change in tree species. The aim of this study was to identify differentially expressed conserved miRNAs in silver birch in vitro samples with differing morphogenic status, and these results confirmed known juvenility-associated miRNAs and identified other miRNA families for further investigation. The rejuvenated and mature in vitro samples were selected according to their propagation ability and morphological properties. However, this assessment was made after approximately 10 months of maintenance in in vitro culture. Assessment of the expression of these miRNAs in a temporal series starting with initiation of mature silver birch explants into tissue culture will identify the miRNAs whose expressions are correlated with rejuvenation processes and which can potentially identify rejuvenated material at an earlier stage than currently possible. The ability to identify rejuvenated material will increase the efficiency of silver birch in vitro propagation, and the results obtained in this study can potentially be transferred to other tree species.

## Figures and Tables

**Figure 1 plants-12-01993-f001:**
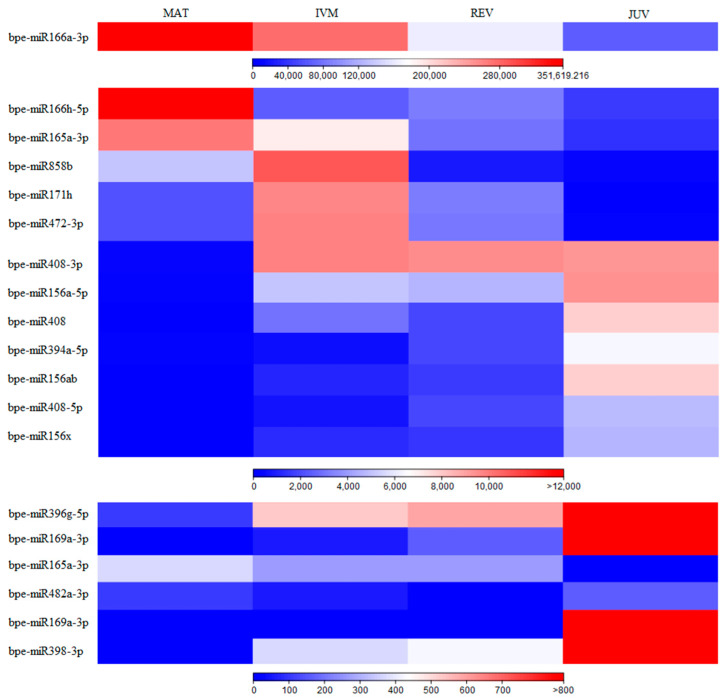
Differentially expressed miRNA between MAT-JUV. Heat map of differentially expressed (with a *p* value ≤ 0.05) miRNAs; the colour bars represent normalised expression values.

**Figure 2 plants-12-01993-f002:**
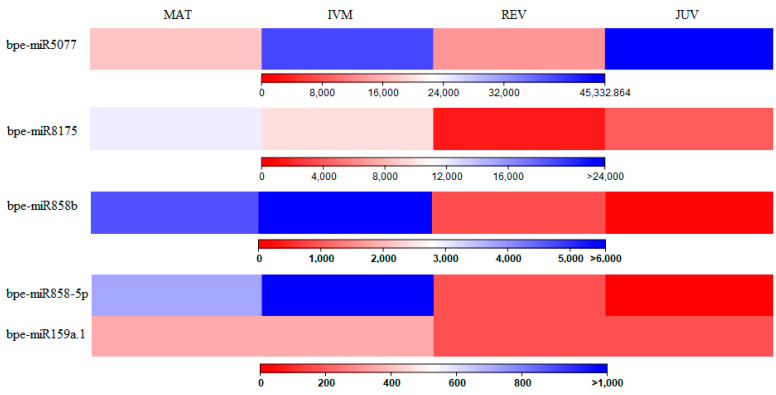
Differentially expressed miRNA between IVM and REV. Heat map of differentially expressed (with a *p* value ≤ 0.05) miRNAs; the colour bars represent normalised expression values.

## Data Availability

The raw sequence reads were deposited in the NCBI database (BioProject ID: PRJNA916462).

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
