# Peer review of "Differentially Expressed Conserved Plant Vegetative Phase-Change-Related microRNAs in Mature and Rejuvenated Silver Birch In Vitro Propagated Tissues"

_plants, 2023, doi:10.3390/plants12101993_

Round 1

Reviewer 1 Report

Dear Authors,

I read the manuscript entitled "Differentially expressed conserved plant vegetative phase-change related microRNAs in mature and rejuvenated silver birch in vitro propagated tissues" with great interest. The identification of miRNAs in non-model species is a topic that needs to be studied intensively because of the role of miRNAs in the regulation of gene expression. Although the concept of the paper is very interesting and has an application aspect in addition to basic knowledge, the manuscript itself and the experimental plan need thorough improvement. 

The material used in the study is described in a way that is difficult to understand (lines 392-396). I assume that two clones VKA and 54-257 were used, from which an in vitro culture was derived, in which some explants were rejuvenated and some were mature. 

It is unclear what these clones (VKA and 54-257) are. Whether they came from a single genotype. What is their history? Why exactly these were chosen for analysis. 

I am also concerned about the number of repeats. For each clone, only two repeats were performed for REV and MAT. A mninimum of three bilogic triplicates is needed for expression analysis. 

So how was the significance of the differences assessed?

There is also no verification of the NGS results using qRT-PCR. Target verification is also lacking. Only in silico analysis was performed, which does not reflect tissue/stadium specificity. miRNAs can regulate many genes, but not all of them need to be expressed in the developmental tissue/stadium under study. Need to verify these results either by transcriptome analysis, degradome sequencing, RLM-RACE, PPM-RACE, or qRT-PCR.

Based on Figures 1 and 2 in the manuscript, there is no significance of differences in levels for several miRNAs. I suggest changing the type of graph because the heat map has too low resolution in this case.

The paper also does not explicitly identify miRNAs that can be used as markers to idnatify MAT and REV explants, nor how to do so. 

The Latin name of a species consists of two parts - the genus name and the species name (species epithet), accompanied at the end by a third element - the name of the author(s) of the name. Latin names should be in italics (populus, oryza)

Best,

M.

The manuscript contains minor linguistic errors. The authors should read it again with care and correct it. 

Author Response

Thank you for the thorough review, below are replies to all specific comments

The material used in the study is described in a way that is difficult to understand (lines 392-396). I assume that two clones VKA and 54-257 were used, from which an in vitro culture was derived, in which some explants were rejuvenated and some were mature. 

It is unclear what these clones (VKA and 54-257) are. Whether they came from a single genotype. What is their history? Why exactly these were chosen for analysis. 

I am also concerned about the number of repeats. For each clone, only two repeats were performed for REV and MAT. A mninimum of three bilogic triplicates is needed for expression analysis. 

  • The methods section was revised to elucidate that the material was obtained from two clones, but for the analysis described in the manuscript, samples were divided according to juvenility/maturity status. Information was also included that they were derived from the silver birch breeding program, but that they were not particularly selected by any criteria other than the morphogenic status of in vitro shoots derived from them. As clarified in the method section, the in vitro samples were divided into groups according to morphogenic status, not taking into account clonal identity. Therefore, four samples were analysed from each group.

So how was the significance of the differences assessed?

            The method section was revised to include criteria for assessment of differential expression of conserved miRNAs

There is also no verification of the NGS results using qRT-PCR. Target verification is also lacking. Only in silico analysis was performed, which does not reflect tissue/stadium specificity. miRNAs can regulate many genes, but not all of them need to be expressed in the developmental tissue/stadium under study. Need to verify these results either by transcriptome analysis, degradome sequencing, RLM-RACE, PPM-RACE, or qRT-PCR.

  • qPCR analysis results from selected differentially expressed miRNS has been included in the manuscript. The methods, results and discussion have been accordingly revised. qPCR results are included as supplementary material. Analysis of the expression of target genes was not in the scope of this manuscript, as commented by the reviewer, miRNAs can regulate many genes, and while this is a research direction we are pursuing, we feel that adequate description of these results will at least double the length of the manuscript.

Based on Figures 1 and 2 in the manuscript, there is no significance of differences in levels for several miRNAs. I suggest changing the type of graph because the heat map has too low resolution in this case.

  • Figures 1 and 2 were revised to make the expression differences more visible.

The paper also does not explicitly identify miRNAs that can be used as markers to idnatify MAT and REV explants, nor how to do so. 

  • The manuscript was revised to remove specific references to markers of juvenility. While this is a long-term goal of our research, additional studies are need to achieve this. This manuscript presents the initial results of differential miRNA expression in silver birch in vitro samples, and they were compared to previous studies of woody tree species. This manuscript provides a basis for further investigations in this field, both for our laboratory, and other researchers.

The Latin name of a species consists of two parts - the genus name and the species name (species epithet), accompanied at the end by a third element - the name of the author(s) of the name. Latin names should be in italics (populus, oryza)

  • Latin species names have been correctly formatted.

Reviewer 2 Report

Authors have documented vegetative phase changes related miRNAs. Typical miRNA sequencing experiments reports known and novel miRNAs. Authors hasn’t reported any novel miRNA in this study.  Also, all the results are based on sequencing data analysis. Authors hasn’t validated any results using RT-PCR or by other mean. Overall manuscript needs bit of additional work.

I have some specific comments.

1.     Please add relevance or importance of work in Abstract. 

2.     Fig1A. Volcano plot. I will suggest adding miRNA names in volcano plot. Just showing the dots is meaningless. Name at least, 10-20 significant miRNAs. Same for Figure 2A.

3.     What happened to novel miRNAs? Why authors not reporting any novel miRNAs in this study.

4.     I suggest including sequencing statistics in manuscript. For example, raw reads generated, clean reads generated, GC content, Q20 and so on in a table.

5.     How target genes selected? How many mismatches allowed? expectation, seed region? Please add detailed information in methodology.

Moderate editing of English language

Author Response

Thank you for the thorough review, below are replies to all specific comments

  1. Please add relevance or importance of work in Abstract. 

  - a sentence about the potential relevance of the obtained results has been added to the abstract.

  1. Fig1A. Volcano plot. I will suggest adding miRNA names in volcano plot. Just showing the dots is meaningless. Name at least, 10-20 significant miRNAs. Same for Figure 2A.

            - Figures 1 and 2 have been revised to remove the volcano plots, and to make them more legible

  1. What happened to novel miRNAs? Why authors not reporting any novel miRNAs in this study.

 - the aim of the study was to identify differentially expressed conserved miRNAs, as stated in the title. We are continuing research efforts, including the identification and characterization of novel miRNAs. However, this requires a lot experimental effort to confirm that small RNA sequences represent miRNAs. The study compared obtained results with previously published reports, increasing knowledge about these processes in woody tree species. As mentioned in the manuscript, most phase change studies have been done in annual species. We think that the results presented in this manuscript extend the knowledge of these processes in woody tree species, particularly with regard to the expression of miRNAs previously reported to have increased expression in mature tissues (e.g. miR172), but which have less clear expression patterns in mature tissues in woody tree species.

  1. I suggest including sequencing statistics in manuscript. For example, raw reads generated, clean reads generated, GC content, Q20 and so on in a table.

            - additional sequencing statistics were included in the text. All sequences have been placed in the NCBI SRA database

  1. How target genes selected? How many mismatches allowed? expectation, seed region? Please add detailed information in methodology.

            - parameters were added to the methods section.

Reviewer 3 Report

This study explored changes in miRNA expression during the transition from the juvenile phase to maturity in birch plants. The findings were presented clearly and are significant in identifying potential markers of both juvenility and maturity in silver birch at an earlier growth stage. These markers can help assess the propagation ability and morphological characteristics of the plants during in vitro propagation. Ultimately, this research may enhance the efficiency of silver birch in vitro propagation and could potentially be applied to other tree species.

Author Response

Thank you for the review, the manuscript has been revised to include issues raised by all reviewers

Round 2

Reviewer 1 Report

It is ok now. I keep my fingers crossed for your further research. I find the topic super interesting

only minor adjustments needed 

Author Response

Thank you for the comment. The language of the manuscript was revised by a native English speaker. 

Reviewer 2 Report

Most of my comments have been answered. But I think the manuscript still needs a little extra work.

1)    Percentage of barcode sequences? Add this information. line 131-132

2)    Change this to “miRNAs showing a fold change in expression ≥ 1.5 or ≤ −1.5 and an FDR-corrected p values < 0.05 were deemed as differentially expressed”. Line 169-170.

3)    The portion of bioinformatics analysis is still not very clear. 

Minor editing of English language required

Author Response

Thank you for the comments and suggestions. All issues raised in the review were addressed. Specific comments are below. The language of the manuscript was revised by a native English speaker.

1)    Percentage of barcode sequences? Add this information. line 131-132

The percentage of barcoded sequences compared to the total number of obtained sequences was added to the results section.

2)    Change this to “miRNAs showing a fold change in expression ≥ 1.5 or ≤ −1.5 and an FDR-corrected p values < 0.05 were deemed as differentially expressed”. Line 169-170.

Changed in results and methods sections

3)    The portion of bioinformatics analysis is still not very clear. 

The methods section describing the bioinformatics analysis was revised and information on parameters used for grouping of isomiRs was added.